# Tergitol Based Decellularization Protocol Improves the Prerequisites for Pulmonary Xenografts: Characterization and Biocompatibility Assessment

**DOI:** 10.3390/polym15040819

**Published:** 2023-02-06

**Authors:** Susanna Tondato, Arianna Moro, Salman Butt, Martina Todesco, Deborah Sandrin, Giulia Borile, Massimo Marchesan, Assunta Fabozzo, Andrea Bagno, Filippo Romanato, Saima Jalil Imran, Gino Gerosa

**Affiliations:** 1Department of Cardiac, Thoracic, Vascular Sciences and Public Health, University of Padua, 35128 Padua, Italy; 2L.I.F.E.L.A.B Program, Consorzio per la Ricerca Sanitaria (CORIS), Veneto Region, 35127 Padua, Italy; 3Department of Industrial Engineering, University of Padua, 35131 Padua, Italy; 4Department of Physics and Astronomy “G. Galilei’’, University of Padua, 35131 Padua, Italy; 5Consultant of Animal and Food Welfare, 35100 Padova, Italy

**Keywords:** decellularization, Tergitol, tissue engineering, RVOTO, congenital heart disease, biocompatibility, mesenchymal stem cells, cytotoxicity

## Abstract

Right ventricle outflow tract obstruction (RVOTO) is a congenital pathological condition that contributes to about 15% of congenital heart diseases. In most cases, the replacement of the right ventricle outflow in pediatric age requires subsequent pulmonary valve replacement in adulthood. The aim of this study was to investigate the extracellular matrix scaffold obtained by decellularization of the porcine pulmonary valve using a new detergent (Tergitol) instead of Triton X-100. The decellularized scaffold was evaluated for the integrity of its extracellular matrix (ECM) structure by testing for its biochemical and mechanical properties, and the cytotoxicity/cytocompatibility of decellularized tissue was assessed using bone marrow-derived mesenchymal stem cells. We concluded that Tergitol could remove the nuclear material efficiently while preserving the structural proteins of the matrix, but without an efficient removal of the alpha-gal antigenic epitope. Therefore, Tergitol can be used as an alternative detergent to replace the Triton X-100.

## 1. Introduction

Congenital heart diseases (CHDs) are among the most life-threatening factors for the pediatric population worldwide. Approximately 0.8% to 1.2% of live births are affected by CHDs, among which Right Ventricle Outflow Tract Obstruction (RVOTO) contributes to 15% of total congenital heart diseases [1,2,3,4,5]. With the remarkable progress in surgical approaches, the percentage of CHD patients who reach adulthood is increasing over time [6]. Due to the lack of an ideal valve substitute, the search for a more efficient replacement intervention with higher durability is not yet completed. Bioprostheses are the most frequently used devices for the replacement of pulmonary valves, albeit their limited durability and residual immunogenicity cause mid to long-term degeneration [7,8]. On the other hand, mechanical prostheses alter the coagulative status, requiring the administration of permanent anticoagulation therapy [9,10]. 

Recently, nanofibers development has made a major contribution to the scope of scaffolds for fabrication that can potentially meet this challenge in combination with an acellular scaffold. Regardless of their fabrication criteria, nanofibers have been used as a scaffold for several applications, e.g., musculoskeletal, heart valve, skin, neural, and vascular tissue engineering [11,12,13]. Tranquillo et al. reported for the first time a tubular heart valve fabricated from two decellularized, engineered tissue tubes connected with absorbable sutures, which met this need, in principle [14].

Tissue-engineered biological heart valve substitutes, with the lowest or no immunogenicity, without the concomitant medical treatment with vast availability and durability, are the ideal to achieve this purpose [15]. Based on the clinical data, homograft is the most favorable, and unfortunately, they have limited availability [16,17]. The substitute, which comes closest to the ideal characteristics, is the decellularized pulmonary xenograft [18]. In general, the decellularized valve should be similar to a scaffold, which maintains the functionality of the native valve and with higher potential for remodeling [19,20]. To obtain an optimal tissue-engineered substitute, the first aim is to find the ideal decellularization protocol that can assure low toxicity and very low damage to tissue structure and composition. 

So far, several decellularization procedures have been reported in the literature [21,22], but most of them are characterized by several limitations, for example, incomplete cell removal, the toxicity of residual detergents, persisting host immune response to the xenograft, thus resulting in a mid-long-term degeneration and calcification [23,24].

The most common detergents applied in the previously reported protocols were SDS, Triton X-100, and Sodium Cholate. Indeed, Triton X-100 has been reported as a hazardous substance for the human endocrine system within the list published by the European Chemical Agency (ECHA) since 4 January 2021 [25].

The specific aim of the present study is to assess the preservation of the ECM scaffold of the decellularized porcine pulmonary valve by using a new eco-friendly detergent, i.e., Tergitol. Our group has reported Tergitol application for the decellularization of the aortic valve [26].

The decellularization protocol includes several treatments, including protease inhibitors’ treatment and the application of Tergitol and sodium cholate detergents in various steps of washings with hyper and hypotonic solutions. The protocol was applied to porcine pulmonary valves. Subsequently, the processed tissues were characterized with regard to their structural and mechanical behavior.

Acellular matrices are prone to cell attachment: this proves the biocompatibility of the decellularized scaffold and the possibility for tissue integration and remodeling in vivo. A well-preserved structure can provide a favorable environment for cell repopulation: the crosstalk between ECM and cells can also promote tissue structural organization, directing cell functions. Understanding the process underlying the crosstalk between cells and ECM will lead to the design of the remodeling process [27,28].

## 2. Methodology

### 2.1. Samples Processing

Fresh porcine hearts (*n* = 26) were obtained from a local slaughterhouse (F.lli Guerriero, Villafranca Padovana, Padova, Italy) following the protocols consistent with EC regulations 1099/2009 regarding animal wealth and protection. After removal from pigs (adults 6–8 months old), hearts were brought to the laboratory within 1–2 h, and then pulmonary valves were dissected and cleaned using isotonic saline. The average 3 cm length of the pulmonary wall was obtained with the leaflets. Samples obtained were frozen at −80 °C.

### 2.2. Decellularization Protocol

The Tergitol-based decellularization protocol was modified from the TRICOL protocol previously reported [19].

Decellularization protocol was performed in an agitation system: treatment with protease inhibitors cocktail (1% *v*/*v*) and DMSO (supplier) (10%) at 4 °C (8 h) was followed by washing with a hypotonic solution (12 h). Subsequently, a second phase with protease inhibitors in combination with 1% Tergitol (12 h) (Sigma Aldrich, Saint Louis, MO, USA) was carried out at room temperature. After further washing, tissues were treated with Tergitol (0.1%) in a hypertonic solution (24 h in two cycles). Thereafter, they were treated for 20 h with sodium cholate (Sigma Aldrich, 4% *v*/*v*). Finally, tissue samples were treated with peracetic acid (Sigma Aldrich, 0.1%) and ethanol 4% (Carlo Erba, Cornaredo, MI, Italy) solutions (90′) for bioburden removal and primary decontamination.

Valves were cut into 8 mm patches with a biopsy puncher and treated with endonuclease enzyme (Benzonase 25 k U, Sigma Aldrich, E1014) at 37 °C for 48 h to complete the removal of nucleic acid residues. Several washing cycles with Phosphate Buffered Saline (PBS, Sigma Aldrich) were performed to remove residues of the enzyme. Samples (Native pulmonary wall (PW Native), Leaflets (LL Native) and Myocardia (MYO Native), Decellularized Pulmonary wall (PW DC), Leaflets (LL DC) and Myocardia (MYO DC))were then embedded in OCT (Tissue-Tek, 4583, Sakura Finetek, Torrance, CA, USA) and stored at −80 °C for histological and IF analysis, while the rest of the tissues were lyophilized for DNA and biochemical analysis. A subgroup of the decellularized valves (*n* = 3) was used for biomechanical tests.

### 2.3. DNA Analysis

Lyophilization of the native (*n* = 4) and decellularized tissues (*n* = 4) was performed with Speed vac SPD130DLX (Thermo Fisher Scientific, Waltham, MA, USA). An amount of 10–12 mg was taken from the pulmonary wall, leaflet, and myocardium tissues and evaluated in duplicate. DNA was extracted according to the manual instructions (DNeasy Blood & Tissue Kit, Qiagen^®^, Redwood City, CA, USA). Quantification of residual nuclear material was performed directly with Nanodrop (Thermo Fischer Scientific). Afterward, extracted samples were analyzed with a Qubit fluorometer by using Qubit™ dsDNA HS Assay Kit (Thermo Fisher Scientific) according to the manufacturer’s instruction.

### 2.4. Histology

OCT-embedded samples from native (*n* = 3) and decellularized tissues (*n* = 3) frozen, as previously described, were frozen and cut (7.0 µm thickness) using a cryostat (NX 70 HOMVPD Cryostar). Haematoxylin and Eosin (Bio-Optica, 04-061010), Masson Trichrome (Bio-Optica, 04-01082), Mallory Trichrome (Bio-Optica 04-020802), Wiegert Van Gieson (Bio-Optica, 04-051802) staining were performed according to the manufacturer’s instructions. Images of stained tissues were obtained at 10× and 20× magnification using an optical microscope EVOSTM XL Core Cell Imaging System (Thermo Fisher Scientific, Waltham, MA, USA).

### 2.5. Immunofluorescence Analysis

The antibodies were applied for the immunological analysis using antibodies indicated in Appendix A to detect structural properties. The primary markers used were Collagen I, Collagen IV, Elastin, and Laminin. Nuclear staining was performed using DAPI (NuBlue Fixed Cell Strain Ready probes reagent, R37606, Thermo Fisher Scientific).

Briefly, the cryosections (7–8 μm thick) were rehydrated with PBS at RT and then fixed in PFA (4% *w*/*v*). After treatment with blocking solution (1% *w/v* of bovine serum albumin, BSA, Sigma), primary antibodies were added to the samples and incubated overnight at 4 °C. Secondary antibodies were added to the sample and incubated at room temperature for 90 min. After secondary antibodies, the sections were incubated with DAPI for 30 min at RT. Slides were then mounted with Mowiol (Mowiol^®^ 4-88 Sigma-Aldrich).

The images were acquired at 10× and 20× magnification with a Leica CTR 6500 fluorescence microscope, and further processing was performed with the LAS AF offline software (Leica Micro-Systems, Wetzlar, Germany). Images of α-Gal epitope were acquired using the confocal microscope Axio Observer LSM 800.

### 2.6. Two-Photon Microscopy

Native and decellularized samples frozen blocks in OCT were cut using a cryostat, measuring 10.0 µm thick cryosections. Tissues were compared using DAPI and Phalloidin 647 staining to evaluate the presence of nuclei. For each sample, 5 different areas were recorded in the two-photon microscope with the same acquisition settings, therefore, directly comparable to each other.

Briefly, an incident wavelength of 800 nm was adopted, and the images were acquired at fixed magnification through the Olympus 25× water immersion objective with 1.05 numerical aperture (1024 × 1024 pixels), averaged over 70 consecutive frames, with a pixel size of 0.43 μm [29].

Qualitative information about the presence of collagen and its distribution in the tissues was obtained with SHG intensity and coherency analysis methods.

### 2.7. Biochemical Analysis

The biochemical analyses included the quantification of hydroxyproline, elastin, and glycosaminoglycans. The protein amount of the decellularized samples was compared to the native tissues.

#### 2.7.1. Hydroxyproline

Native (*n* = 3) and decellularized (*n* = 3) samples of the pulmonary wall, leaflet, and myocardium (3–5 mg each) were used for this quantification. Hydroxyproline extraction was performed using a hydroxyproline assay kit (MAK008, Sigma-Aldrich) [30]. Absorbance was measured at 560 nm with a spectrophotometer reader (SPARK).

#### 2.7.2. Elastin

Elastin quantification was performed with the colorimetric analysis of The Fastin™ Elastin Assay protocol (Biocolor F4000, county Antrim, UK). Elastin content was measured in the pulmonary wall, leaflet, and myocardium of native (*n* = 3) and decellularized (*n* = 3) pulmonary valves (3–5 mg each). Absorbance was measured with a SPARK spectrophotometer at a wavelength of 513 nm.

#### 2.7.3. Glycosaminoglycans

GAGs quantification was performed by colorimetric analysis using the Blyscan™ Glycosaminoglycan Assay (Biocolor, B1000, County Antrim, UK) protocol [31]. Native (*n* = 5) and decellularized (*n* = 5) valves were evaluated in all the components: pulmonary wall, leaflet, and myocardium (3–5 mg each). Absorbance was measured at 656 nm with the SPARK spectrophotometer.

### 2.8. Biomechanical Tests

Uniaxial tensile tests were performed on the pulmonary arterial wall and leaflets to biomechanically assess the effects of the decellularization procedure. Three sets of porcine pulmonary heart valves, native and decellularized, were analyzed.

The pulmonary valve was opened with a cutter, and 9 dog-bone-shaped samples were taken from each valve. All specimens were cut with a custom-made cutter with a gauge length of 5 mm and a width of 2 mm in the central section. The leaflets were isolated and cut, and one sample was obtained from each leaflet. Six samples were obtained from the pulmonary wall, three in the circumferential direction and three in the longitudinal direction. The sample thus obtained from both were uniaxially tested but can lead to assessment anisotropy [32].

Primarily, the thickness of each tissue sample was measured using Mitutoyo digital Caliber model ID-C112XB (Aurora, IL, USA), and later, the test for the uniaxial tensile property was performed with a customized apparatus (IRS, Padova, Italy) aided by a LabVIEW software (National Instruments, Austin, TX, USA). Tests were carried out as previously described [26,27]. Briefly, the samples were preloaded to 0.1 N at room temperature, then stretched until ruptured at the rate of 0.2 mm/s. Displacement and load were recorded with a sampling frequency of 1000 Hz. Acquired data were analyzed using an in-house developed Matlab^®^ script: for each sample, engineering stress σ (MPa) and strain ε (%) was calculated as the applied load divided by the initial cross-sectional area and the current length divided by the initial length, respectively. Ultimate Tensile Strength (UTS) and Failure Strain (FS) were also calculated in terms of the maximum strength and elongation of each sample.

In order to better characterize the non-linear stress-strain curves obtained from each sample, two tensile moduli, E1 and E2, were calculated as the slope of the linear portion of the curve between 0–10% (elastin region) and 60–100% deformation (collagen region), respectively [26,28]. A *t*-test (GraphPad Prism Software, San Diego, CA, USA) was applied to calculate the statistical analysis. Significance was set at *p* < 0.05.

### 2.9. Scanning Electron Microscopy Analysis

To evaluate the effect of decellularization on pulmonary valves, a scanning electron microscope (SEM) was used [33]. One sample from the pulmonary wall was collected at each step of the procedure (after Tergitol, sodium cholate, Benzonase^®^ and one after cell seeding) and compared to the native tissue. Photographs of each patch were taken at 200×, 3000×, and 8000× magnification.

### 2.10. Sterility Assessment

The sterilization of decellularized tissues was performed following the guidelines of the European Pharmacopoeia 2019 [34]. The decontamination of decellularized samples was performed in two phases: first, samples were treated with 70% ethanol for 30 min at RT. Afterward, they were treated with a cocktail of antibiotics and antimycotics (including vancomycin hydrochloride (50 mg/L, SBR00001, Merck), gentamicin (8 mg/L, G1397, Merck), cefoxitin (240 mg/L, C0688000, Merck) and amphotericin B (25 mg/L, A9528, Merck)) at 37 °C for 24 hr.

For the sterility assessment, a thioglycolate medium, (T9032, Sigma-Aldrich) and soya-bean casein digest medium, (22092, Sigma-Aldrich) were used, and turbidity was observed over different time intervals from Day1–14. The patches (8 mm) from native, decellularized and decellularised, and decontaminated samples of the pulmonary wall, leaflets, and myocardium (each in duplicate) were immersed into media to detect the growth of aerobic/anaerobic bacteria and fungal growth. Samples in Thioglycolate medium were incubated at 35 °C in the oven, and those immersed in Soya-bean casein digest medium were kept at RT for 14 days. Images were taken during incubation at time intervals of 48 hrs.

### 2.11. Cytotoxicity/Cytocompatibility

hMSC-BM (12974, PromoCell) were cultured in ready-to-use specific media (Mesenchymal Stem Cell Growth medium, C-28009, PromoCell). Cells were expanded and passaged up in an incubator with 5% CO_2_ at 37 °C and 95% humidity. Sterilized tissue patches were equilibrated with the MSC media for 24 hrs. prior to cell culture. Cells were harvested and resuspended in media with a density of 1 × 10^6^ cells/mL aseptically. Approximately 30,000 cells were on the top of each tissue. The proliferation/toxicity analysis was performed at each time point (24 h, day 7, day 14 after seeding). The analysis was evaluated by the WST-1 assay, Live/dead staining, and DNA quantification.

#### 2.11.1. Live and Dead Staining

LIVE and DEAD^®^ Viability/Cytotoxicity Kit (L3224 Thermo Fisher Scientific) was used. Staining was performed according to the manufacturer’s instructions.

#### 2.11.2. DNA Quantification

Quantification of DNA was performed as previously described in the characterization of the decellularized valve: Nanodrop (Quick-Start Protocol DNeasy Blood & Tissue Kit, QIAGEN) and Qubit assay (Qubit™ dsDNA HS Assay Kit, Thermo Fisher Scientific) were used. DNA content was expressed in ng/mg of wet tissue.

#### 2.11.3. WST-1 Assay

WST-1 solution was made of WST-1 (5%) and mesenchymal cells media., and a volume of 300 µL of this solution was added to each patch and incubated at 37 °C for two hours, according to the supplier’s instructions. Absorbance was measured at 450 nm.

#### 2.11.4. SEM Analysis

Decellularized and decontaminated pulmonary tissue patches (*n* = 3) were imaged using SEM after 14 days of hMSC-BM culture.

### 2.12. Statistical Analysis

All data were expressed as mean ± SD. One-way ANOVA was performed with GraphPad Prism 8 to compare the groups of experiments, and multiple comparisons with each control group were performed. Differences were considered statistically significant when *p* < 0.05.

## 3. Results

### 3.1. Morphology

The decellularized porcine valves appeared white, whereas the myocardium exhibited a light brown compared to the native state (Figure 1b,c). Lyophilized samples were used for DNA quantification and biochemical assays. Freshly decellularized tissues were used along with native aortic valves for biomechanical tests. OCT-embedded tissue punches (8 mm) were frozen in liquid nitrogen for histology and immunological analysis.

### 3.2. DNA Quantification

DNA quantification showed a significant reduction in the content of double-strand DNA in treated tissues both with Nanodrop and Qubit assays. Nanodrop showed a decrease in DNA content to 3.54% for the pulmonary wall, 1.47% for the pulmonary leaflet, and 4.37% for the myocardium. Qubit showed a similar reduction in terms of percentage, with a residual DNA of 2.77% for the pulmonary wall, 3.53% for the leaflet, and 1.26% for the myocardium (Figure 1d,e).

### 3.3. Histological Analysis

Cell nuclei are very well represented in native tissues, while they are completely absent in decellularized samples. Figure 2a illustrates the results of Haematoxylin and Eosin staining.

Collagen seemed well preserved and maintained its distribution in all decellularized tissues as it is likely to observe in three different connective stainings using Masson trichrome (Figure 2b), Mallory trichrome (Figure 2c), Weigert Van Gieson (Figure 2d). Mallory trichrome staining (Figure 2c) and Wiegert Van Gieson staining (Figure 2d) showed a reduction in elastin content. The three-layered organization of the pulmonary wall seemed preserved in all histological samples.

### 3.4. Immunofluorescence

DAPI nuclear staining confirmed the effective removal of cells in the decellularized leaflet, pulmonary wall, and myocardium with immunofluorescence and two-photon microscopy (Figure 3). Immunofluorescence showed great preservation of the major structural proteins (Elastin, Collagen I and IV, and Laminin) in the extracellular matrix. The structural architecture was well preserved in the leaflet (Figure 3b), myocardium (Figure 3c), and in all three components of the pulmonary wall: adventitia, media, and intima (Figure 3a). α-Gal persisted in decellularized tissues, with only a slight reduction in the corresponding signal (Appendix A).

### 3.5. Two-Photon Microscopies

To deepen the investigation of structural proteins, with a peculiar interest in collagen, we evaluated Collagen I in a semi-quantitative approach with SHG imaging. Compared to immunofluorescence staining, which requires the processing of the tissues, SHG offers a label-free approach to acquire Collagen I signal proportional to the protein content. Moreover, samples were stained with DAPI.

In Figure 4a, representative images of native and decellularized tissues from the pulmonary wall, leaflet, and myocardium are shown. DAPI signal is completely absent in decellularized portions. This observation further confirms the effective nuclei removal after the decellularization protocols. First-order analysis of the SHG signal is focused on the average intensity that is proportional to protein content. We observed a decrease in signal intensity in the pulmonary wall and leaflets after decellularization, while no changes appeared in myocardium samples (Figure 4b). In line with this, a re-arrangement of the collagen fibers resulted from coherency analysis for the pulmonary wall and leaflets. Again, no differences have been observed in the myocardium comparing native and decellularized tissue (Figure 4c).

### 3.6. Biochemical Analysis

The amount of hydroxyproline in the decellularized valves was higher than the native ones for the pulmonary wall, leaflet, and myocardium (Figure 5a). However, statistically, the quantity of collagen is within the non-significant range except in myocardial tissue. Elastin was shown to be sustained in the pulmonary wall and leaflet but significantly decreased in the myocardium (Figure 5b). GAGs amount seemed to be significantly reduced in all three tissues of the pulmonary scaffold: pulmonary wall, leaflet, and myocardium (Figure 5c).

### 3.7. Biomechanical Results

Examples of the stress-strain curves obtained from the uniaxial tensile tests performed on pulmonary valves, along both the circumferential and longitudinal directions, and leaflets are depicted in Figure 6a. It is likely to check that the leaflets exhibit a response to load different from the valve wall, whereas the decellularization procedure seems not to cause appreciable changes in the mechanical behavior of the investigated tissues. Decellularized samples are less thick than native ones in both pulmonary walls and leaflets (Figure 6b) but without statistically significant differences.

With respect to UTS and FS values (Figure 6b), the decellularization causes a significant increase in the mechanical strength of the leaflets (*p* = 0.003) and the pulmonary wall along the circumferential direction (*p* = 0.03); there is a non-significant decrease in UTS values along the longitudinal direction in decellularized samples with respect to the native ones. The maximum elongation, FS, significantly increases in the circumferential direction (*p* = 0.02) while it decreases in the longitudinal direction and the leaflets.

With regard to Young’s moduli E1 and E2 (Figure 6b), which are characteristic of the elastin and collagen parts of the stress-strain curves, the first one reaches lower values than the second, in accordance with the literature [30]. After decellularization, E1 significantly decreases along the circumferential direction of the valve wall (*p* = 0.002), while it increases in the longitudinal direction (*p* = 0.02) and leaflets (*p* = 0.02). The same trend is observed for E2: there is a significant decrease in stiffness along the circumferential direction (*p* = 0.0009), while there is an increase in the longitudinal direction (but non-significant) and leaflets (*p* = 0.004).

### 3.8. Scanning Electron Microscopy (SEM)

SEM analysis showed a slight reduction in thickness in decellularized tissue compared to the native one. With the loss of the quaternary structure of the pulmonary wall, the surface appears smoother and more homogeneous, but the overall structure is preserved.

Samples from the pulmonary root were analyzed at different steps along the decellularization process: native, after treatment with Tergitol, sodium cholate, and Benzonase, and after 14 days from cell seeding with hMSCs-BM (Figure 7). Images of native tissue showed the presence of cells and the typical structure of collagen fibers inside the matrix. After Tergitol treatment, cells appear in a lower amount or are completely absent, while collagen structure is slightly disrupted. After sodium cholate treatment, pictures show the presence of cellular debris while collagen fibers are maintained: less tissue disruption is caused during this step. After Benzonase, nuclear components are completely absent.

In native tissue samples (Appendix A), high turbidity is clearly visible within 24–72 hrs. Decellularized samples were shown to be turbid on day 7. In contrast, sterilized tissue patches were shown to be without turbidity until the end of the test, i.e., day 14.

### 3.9. Cytotoxicity/Cytocompatibility Tests

Regarding cytocompatibility, pulmonary wall, and leaflet exhibited different results. From the beginning, WST-1 test demonstrated that the pulmonary wall showed lower values of formazan optical density than the leaflet, and after 72 h, these values have been progressively decreasing (Figure 8). Similar results were obtained by DNA quantification (Figure 8) and confirmed by LIVE/DEAD staining (Figure 9), where the number of live cells decreased progressively from day 7 to day 14. In the leaflet, WST-1 assay showed high optical density from 24 h, and these values grew progressively until stabilization of the growth curve between day 7 and day 14. Similarly, the DNA amounts increased over time: these results were confirmed by live and dead staining.

## 4. Discussion

In the present study, a newly reported detergent (Tergitol) was used to decellularize the porcine pulmonary valve. The Tergitol-based protocol proved to be effective in removing cellular components and debris from the native tissue.

Histological evaluation allowed for assessing the complete removal of nuclei and the preservation of the extracellular matrix as demonstrated by three different stainings that are specific for the connective tissue (e.g., Massons’ trichrome, Mallory trichrome, Weigert Van Gieson). DAPI confirmed the absence of nuclei in the decellularized tissues both with immunofluorescence and with two-photon microscopy. Immunofluorescence showed no difference in the expression of collagen I, collagen IV, laminin, and elastin. Although the wall thickness of the pulmonary artery was slightly reduced after decellularization, SEM analysis revealed that its surface was not altered by the detergents used. The quaternary structure of matrix proteins, however, was damaged, with a visible reduction in the organization of the fibers and improved smoothness of the pulmonary wall surface.

Detection of α-Gal epitope was performed in two ways: a signal derived from the anti-α-Gal epitope antibody and a signal derived from isolectin binding to the antigen were measured. Our results showed that the Tergitol-based decellularization procedure is able to reduce the amount of α-Gal epitope but not to remove it completely. Ongoing investigations are aimed at exploring other strategies for the effective removal of this antigenic epitope before applying the scaffold for implantation in the animal model [33].

Uniaxial tensile tests showed no substantial differences in decellularized and native pulmonary valve tissues, so it is possible to hypothesize that the decellularized scaffold maintains its mechanical properties and functionality.

DNA quantification was performed using two different assays, Nanodrop and Qubit: in both cases, the proposed decellularization protocol was demonstrated to be effective in removing cells nuclei and nucleic acids: DNA amount was lower than 50 ng/mg in the decellularized pulmonary wall, leaflet, and myocardium.

The higher level of hydroxyproline detected in decellularized tissues compared with native ones can be explained by two hypotheses: first, collagen can be masked by the cells in native tissue so that the assay cannot detect it. SEM analysis showed that the decellularized tissue possesses more disrupted loose fibers, while native tissue is more compact: the compactness of the native tissue can make it difficult to digest, resulting in a lower collagen extraction. The second hypothesis suggests that, with the same weight, the decellularized tissue has a much higher concentration of the ECM than the tissue-containing cells. However, this test quantitatively showed that collagen is well preserved after decellularization, confirming qualitative results.

Elastin quantification proved a slight decrease in the amount of this protein, but it is not statistically significant in pulmonary walls and leaflets. In the myocardium, the loss of elastin is significant, but it can be neglected since the myocardium is not relevant to the functionality of the pulmonary valve.

Decellularization induced a decrease in the level of GAGs, which are the binding sites of a large number of growth factors and chemokines. GAGs localize the mediators to specific sites in tissues and influence their stability and function [34]. For this reason, the loss of these molecules can reduce the ability of cells to migrate and colonize the scaffold upon implantation. However, GAGs are also deemed to play a role in inflammation and immune response: thus, their reduction in decellularized pulmonary tissues can be considered advantageous to prevent calcification and immune response [34,35].

Biocompatibility tests showed excellent cell growth on the leaflets but not on the pulmonary wall. Biocompatibility was tested qualitatively with WST-1 proliferation index assay and live and dead staining and quantitatively using DNA quantification. All tests showed that mesenchymal cells did not grow on the pulmonary wall, and their amount decreased over time. Indeed, mesenchymal cells did not find a favorable environment in the pulmonary wall: it can be due to inadequate growth stimulation or the absence of specific growth factors for these cells. Furthermore, the thickness of the wall can cause some detergent residues to remain trapped in the tissue: these residues could be toxic to mesenchymal cells. However, previous studies demonstrated that decellularized allografts implanted in the animal model showed an increase in DNA content in the wall comparable to the original root 15 months after implantation [17].

The leaflets showed a progressive growth of mesenchymal cells from both a qualitative and quantitative point of view. The leaflet is much thinner than the pulmonary wall, and this may partly contribute to cell growth since potentially cytotoxic elements were more easily removed.

## 5. Conclusions

In our study, we observed that, with the application of Tergitol, decellularization protocol proved to be efficient in removing cells and debris from the porcine pulmonary root, leaving the extracellular matrix well maintained in the anatomical and histological organization. Uniaxial tensile tests let us hypothesize that functionality of the valve is maintained too.

The Tergitol modified protocol, therefore, allows obtaining a pulmonary valve scaffold that could mirror the characteristics of an ideal valve substitute. However, the persistence of αGal antigen in tissues requires some specific treatments, as it is not sufficient to remove these xenoantigens completely.

Cytotoxic and cytocompatibility tests showed that leaflets might be an optimal substrate for the growth and proliferation of bone marrow mesenchymal cells; however, limited growth in the pulmonary wall requires further in-depth studies to assess the possible reasons involved in this cellular behavior, and if needed, further, surface modifications in the tissue can possibly improve the cell tissue interaction. From the present results, it can be suggested to implant the decellularized valve in an animal model to evaluate both the in vivo functionality and hemodynamic properties of the scaffold.

## Figures and Tables

**Figure 1 polymers-15-00819-f001:**
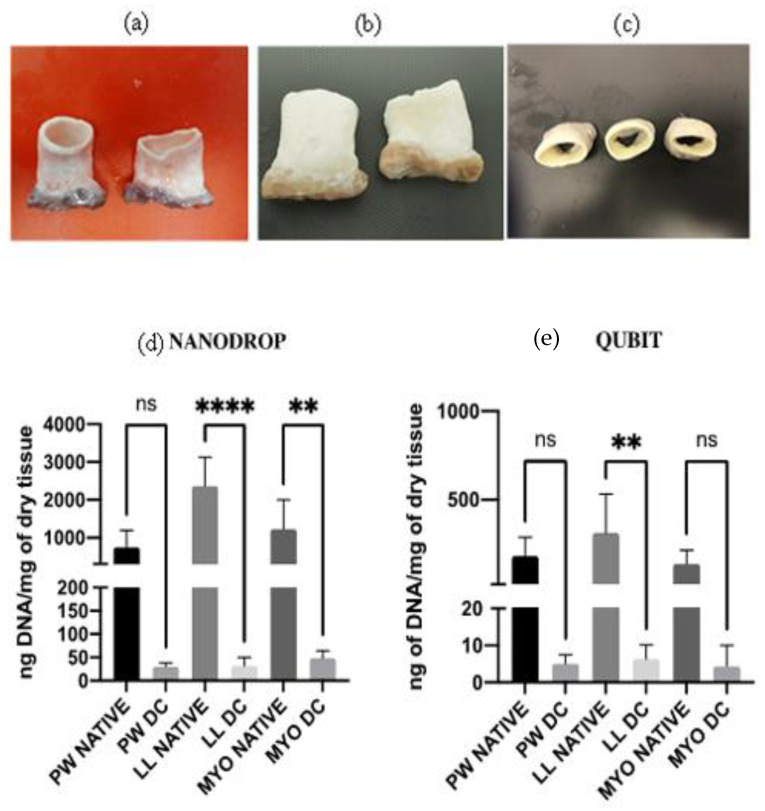
Macroscopic features of (**a**) native and (**b**,**c**) decellularized porcine pulmonary valves. DNA quantification (ng/mg dry tissues) of native and decellularized porcine pulmonary tissues performed with (**d**) Nanodrop and (**e**) QUBIT assays. Data were statistically compared by one-way ANOVA (*p* < 0.05). Decellularized tissue showed in both the analysis and DNA content below 50 ng/mg, which is the threshold to define a well decellularized tissue (values (** *p* < 0.001, **** *p* < 0.00001)).

**Figure 2 polymers-15-00819-f002:**
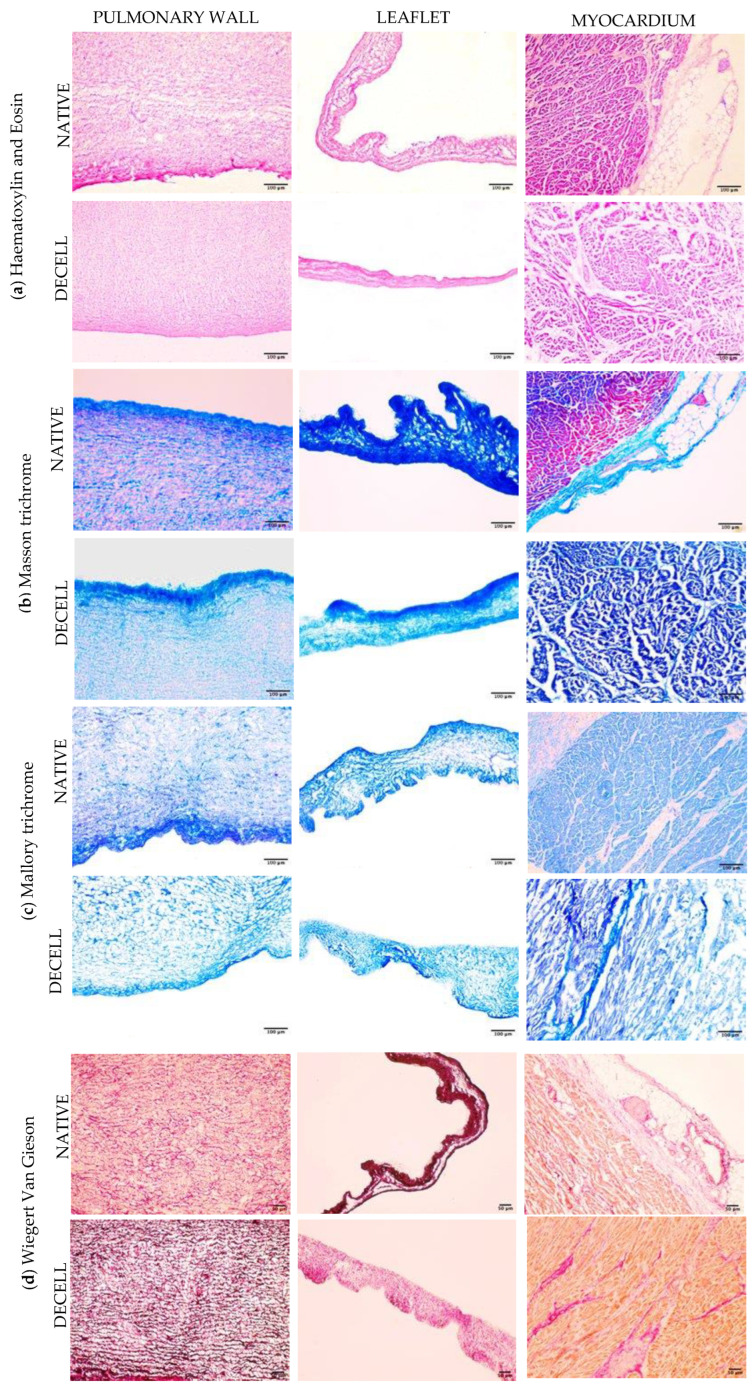
Histology of native and decellularized pulmonary wall, leaflet, and myocardium. Staining was performed using: Hematoxylin and Eosin (**a**), Masson trichrome (**b**), Mallory trichrome (**c**), and Wiegert van Gieson (**d**). Cryosection thickness = 7 µm. Images were obtained at 20× magnification. Scale bar = 50 and 100 µm.

**Figure 3 polymers-15-00819-f003:**
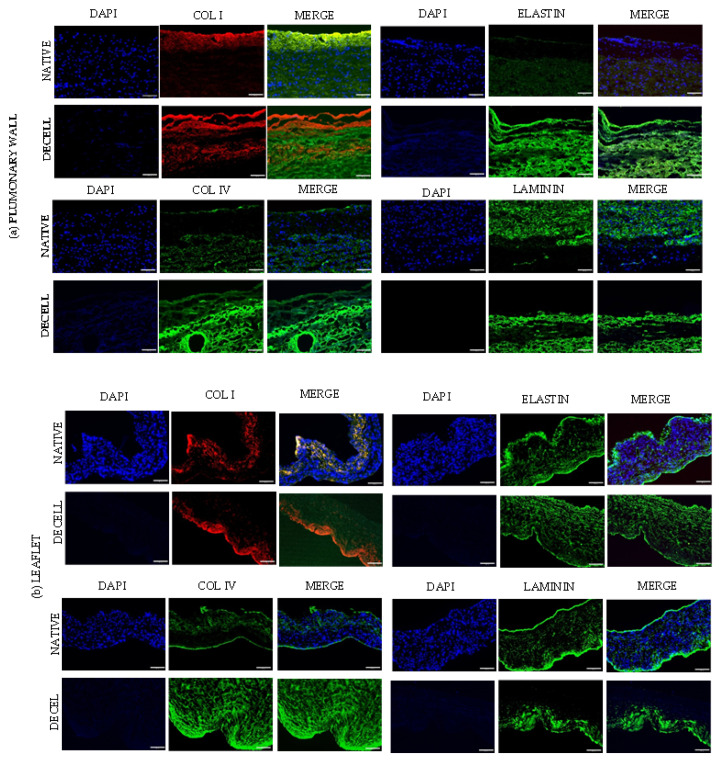
Immunofluorescence analysis of native and decellularized (**a**) pulmonary wall, (**b**) leaflet, (**c**) myocardium. PFA fixed cryosection (7 um) stained with anti-collagen I (Col I, red), collagen IV(COL IV), elastin and laminin (green) antibodies, and DAPI (blue). Imaging was performed on a Leica imaging system at 20× magnification. Scale bar 50 um.

**Figure 4 polymers-15-00819-f004:**
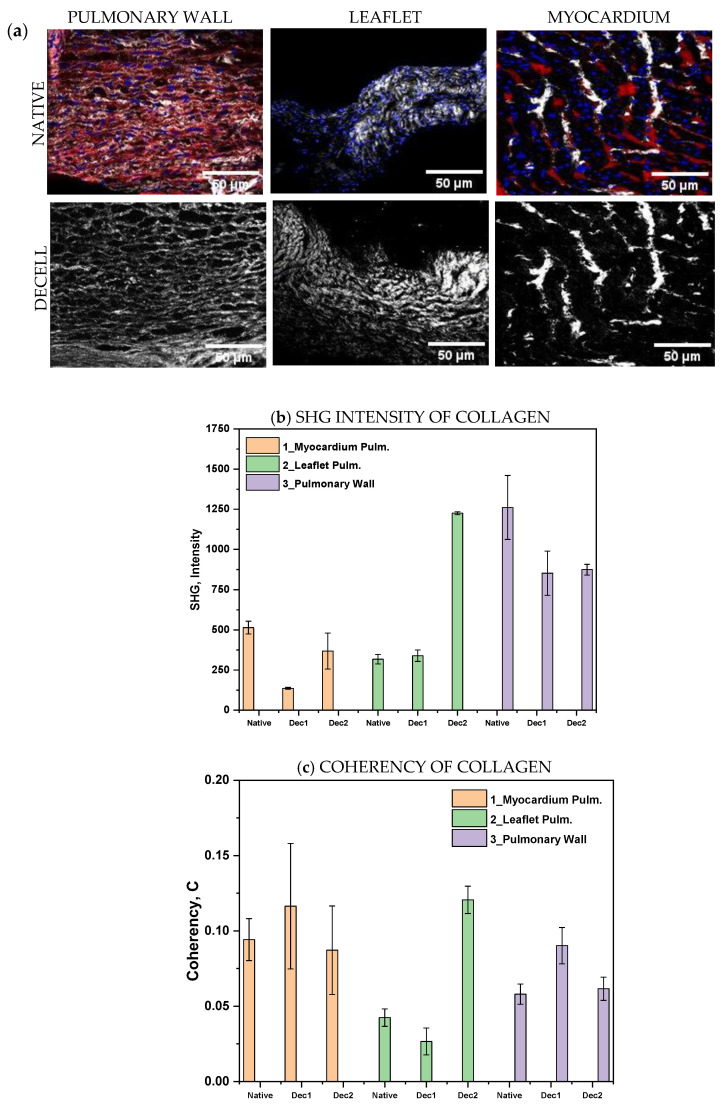
(**a**) Two-photon microscopy images of native and decellularized pulmonary wall, leaflet, and myocardium. (**b**) SGH intensity and (**c**) coherency of collagen. SGH = Second Hormonic Generation, Dec1 and Dec2 are mentioned for two independent decellularization experiments, 1 and 2, to compare.

**Figure 5 polymers-15-00819-f005:**
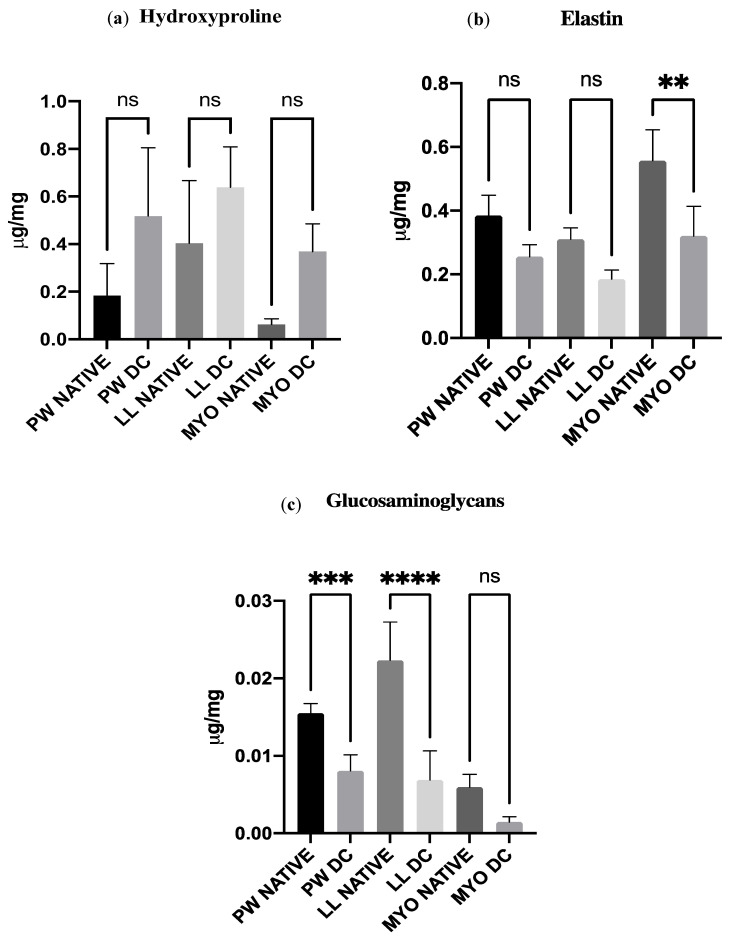
Biochemical analysis of (**a**) glycosaminoglycans, (**b**) elastin and (**c**) hydroxyproline shown. Native (*n* = 3) and decellularized (*n* = 3) samples were used for the analysis. Data were analyzed by one-way ANOVA (** *p* < 0.001, *** *p* < 0.0001, **** *p* < 0.00001). There was a non-significant (ns) difference between native and corresponding decellularized tissues. Increasing values of hydroxyproline are not statistically significant.

**Figure 6 polymers-15-00819-f006:**
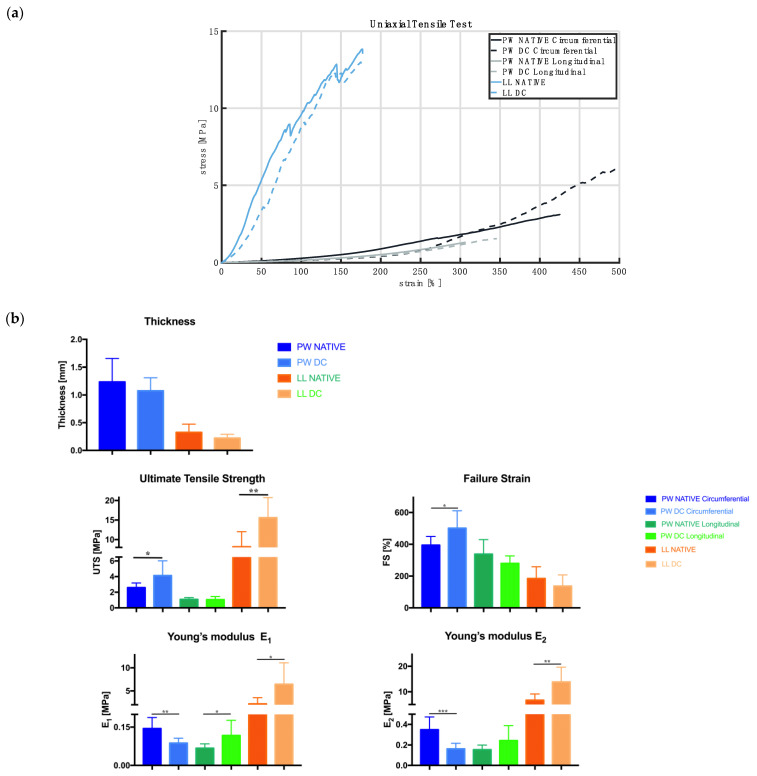
Biomechanical data: (**a**) example of the stress-strain curves comparing the different behavior of native and decellularized samples taken from the pulmonary wall, tested along the circumferential and longitudinal directions, and leaflets. (**b**) Parameters for the detection of mechanical behavior using sample thickness, Ultimate Tensile Strength (UTS), Failure Strength (FS), Young’s Modulus E1 and E2 values (* *p* < 0.05, ** *p* < 0.001, *** *p* < 0.0001).

**Figure 7 polymers-15-00819-f007:**
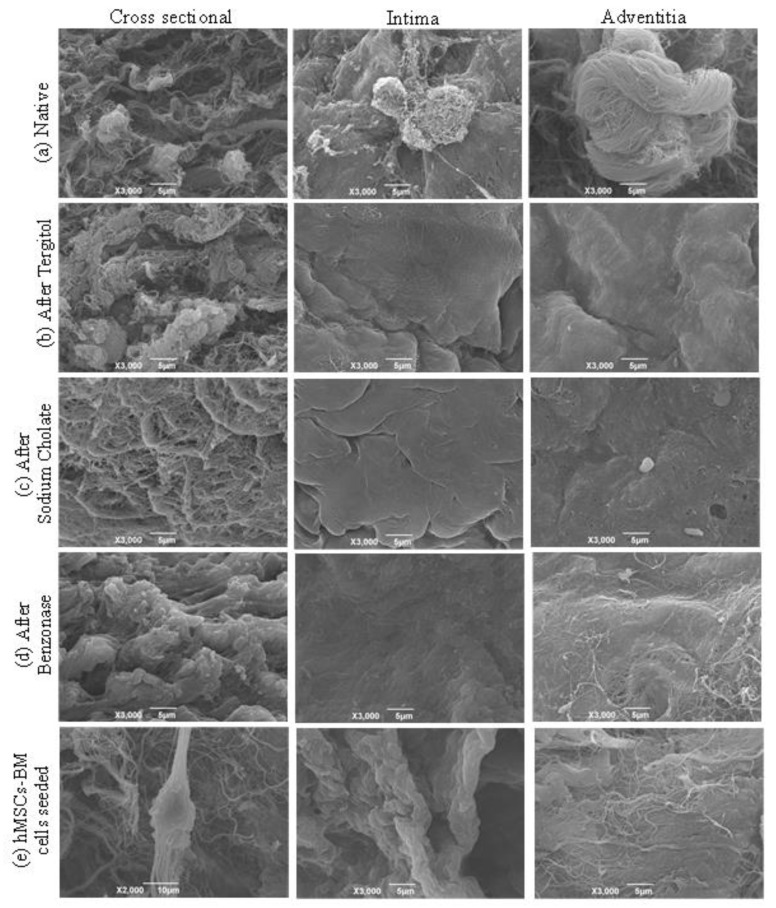
SEM imaging showed the effects of the most important step of the decellularization protocol in the surface of the pulmonary wall. (**a**) native tissues, (**b**) after Tergitol step, (**c**) after sodium cholate step, (**d**) after benzonase treatment, (**e**) after mesenchymal stem cells seeding. 3.9. Sterility Assessment.

**Figure 8 polymers-15-00819-f008:**
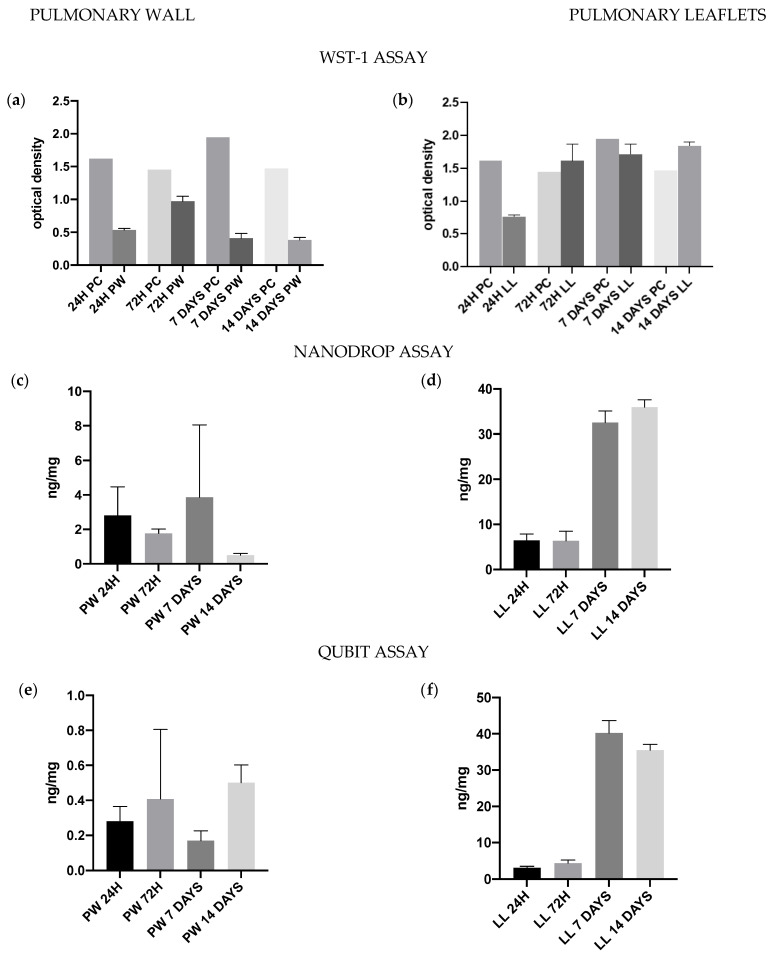
WST-1 proliferation assay for the pulmonary wall (**a**) and leaflet (**b**). the assay was performed at different time points from mesenchymal cell seeding. PW = pulmonary wall, LL = leaflet, PC = positive control (only cells). WST-1 results confirmed the ones of DNA quantification and Live and Dead staining. DNA quantification at different time points from cell seeding. The analysis was performed for the pulmonary wall (**c**,**e**) and leaflet (**d**,**f**) with both Nanodrop (**c**,**d**) and QUBIT (**e**,**f**) assays. The pulmonary wall showed a decrease in DNA content during the 14 days, confirming the results of Live and Dead staining and WST-1 assays. Leaflet showed a great increase in DNA content, especially between 72 h and 7 days time points.

**Figure 9 polymers-15-00819-f009:**
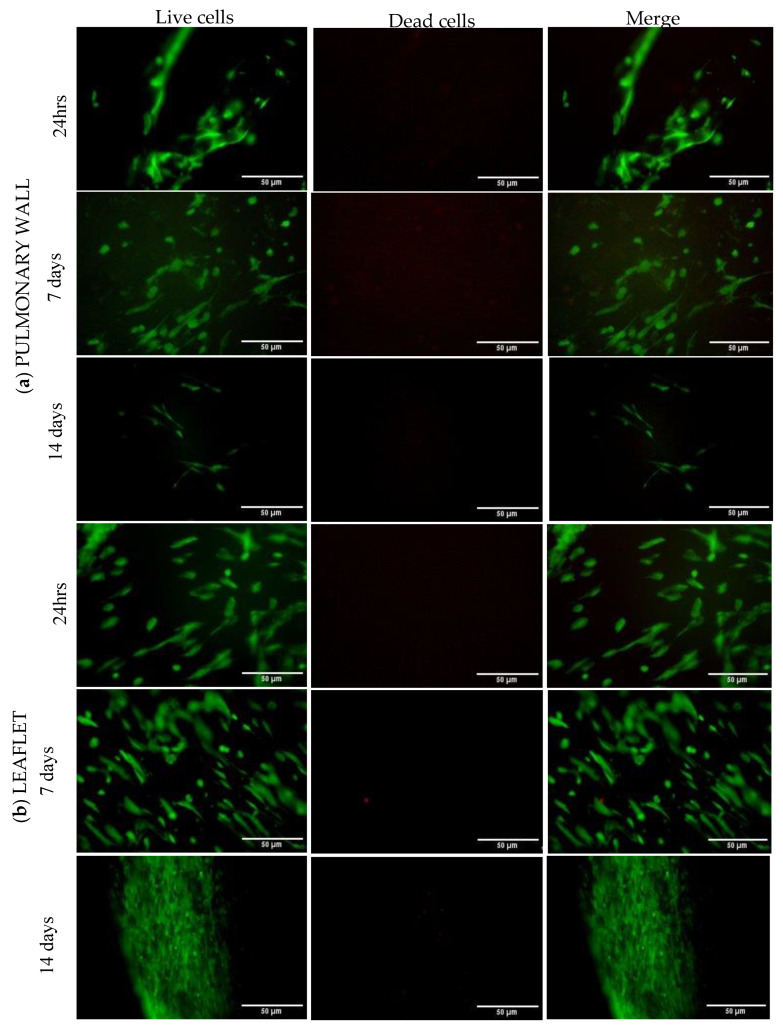
Live and dead staining for (**a**) pulmonary wall and (**b**) leaflet at different timepoint from mesenchymal cell seeding. Live cells are shown in green, while dead cells are in red. The pulmonary wall showed a progressive decrease in the number of cells for 14 days, while the leaflet showed an evident increase in them over time.

## Data Availability

Not applicable.

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
