# Peer review of "Tergitol Based Decellularization Protocol Improves the Prerequisites for Pulmonary Xenografts: Characterization and Biocompatibility Assessment"

_polymers, 2023, doi:10.3390/polym15040819_

Round 1
Reviewer 1 Report (Previous Reviewer 1)
the authors have addressed all my comments.
Author Response
Dear Prof.
I am very thankful to your kind support.
Regards,
Reviewer 2 Report (New Reviewer)
Nowadays, the high percentage of RVOTO as the congenital heart pathology demands a pulmonary valve substitution with an ultimate tolerance. To provide this procedure, a series of homo- and xeno-transplantats are widely implemented on the base of decellularization protocol to avoid trombogenicity, immunogenic response, calcification, and other deleterious phenomena. For the porcine pulmonary valve, the anatomical organization, as well as the functionality of the extracellular matrix, persist in good condition after a cell elimination prescript based on a novel detergent, Tergitol. Additionally, in the presented submission, cytotoxic and cytocompatibility trials demonstrate that the leaflet could be a relevant substrate for the growth and proliferation of bone marrow mesenchymal cells.
With appropriate terminology and reasonable argumentation, the manuscript topic promotes the trend of novel transplant design. On the whole, the manuscript abstract reflects the general issues of the paper. The literature cited is quite relevant to this study and the illustrations (the column schemes and the figures) are executed in an unambiguous and accurate manner with a coherent interpretation. The results of the statistical analysis are clearly presented in a quite reasonable form and have corresponding support in the methodical section.
Nevertheless, the coherent sequence of findings has certain weaknesses that are pointed out below
The part of introductory phrases could be omitted to shorten the Abstract. The reviewer recommends persistently avoiding the part of the Abstract after the phrase “requires subsequent pulmonary valve replacement in adulthood.” without the expanded discussion that should be located in the introductory part. Instead of this analysis mentioning, it is worth enhancing the abstract content by the addition of the principal issues of the submission such as the uniaxial tensile test conclusion, the comparison of Tergitol with the replaced detergent (Triton X-100), perspectives of the author activity like in Conclusion partition, and the other items optionally.
In the Introduction, it is quite appropriate to include a short paragraph devoted to the progress of valve design that has used polymer biomaterials, e.g. recent investigation on biomaterial-based tissue-engineered heart valve https://doi.org/10.1016/j.matpr.2020.07.712; Materials and manufacturing perspectives in engineering heart valves:
a review https://doi.org/10.1016/j.mtbio.2019.100038; electrospun, nanofiber scaffolds https://doi.org/10.1016/j.biopha.2016.10.058; nanofibrous bioengineered heart valve—Application in pediatric medicine, etc. This piece of the survey will be in the line with the Polymers scope.
L26: The abbreviation of the matrix (ECM) should be disclosed immediately after the mentioning
L66: It is worth formulating as “protease inhibitors' treatment”
L74: The fragment of the expression “The substitute, which comeS” looks like a typo
L90: It is rather 8 h than hrs and farther in page 3, L91,92: 12 h; as well as in LL: 94,95
L98: to convey the dynamic character of the procedure it is better to express the action as “cut into”
LL694 – 698. Here, the sentence is formulated as a very complicated fragment and should be simplified e.g. via dividing into two separated sentences expressing the same ideas.
Summarizing the Reviewer's opinion, it should recommend proceeding with this manuscript to the following Editorial/Publishing performance after making the above minor corrections.
Author Response
Dear Prof.
I am thankful on this occasion to find your useful suggestions to improve the manuscript. Please find our responses in the attached file.
best regards,
Saima

Reviewer 3 Report (New Reviewer)
The paper is very well written and presented. The idea is interesting and everyone contributes to scientific research. Kudos for the paper. I have a suggestion to add more relevant recent papers in this field to the list of references.
Author Response
Dear Prof.
Thank you very much for your time to review our work. Please fid our responses in attached file.
Best regards,
Saima

This manuscript is a resubmission of an earlier submission. The following is a list of the peer review reports and author responses from that submission.
Round 1
Reviewer 1 Report
This manuscript by Susanna Tondato et al. reported using Tergitol as a new decellularization technique to fabricate valve substitute. The authors did extensive characterization on how their decellularization technique affects the DNA content, extracellular matrix structure and component, mechanical property, and biocompatibility of the final product. The data collectively showed that this new protocol could be used as a potential decellularization technique. However, the data failed to support that this technique is advantageous over current SDS or Triton based decellularization due to lack of a control decellularization group. Therefore, the conclusion “Tergitol can be used as an alternative detergent to replace the Triton X-100.” In addition, the following issues need to be addressed before the acceptance for publication.
Major issues:
1. A Triton x-100 control or SDS group is needed.
2. At page 3 paragraph 2, please added the duration of each treatment. How long was the 1% Tergitol treatment, sodium cholate treatment, peracetic treatment?
3. Figure 1, the panel label e is missing in the figure. In panel d and e, seems like the author only did paired two group comparison? However, the legend says statistic was done by one-way ANOVA. Please confirm this.
4. Figure 2, the scale bar is not readable and I can barely see the structure in the histology images. Please include higher resolution images.
5. Figure 3, DAPI staining in elastin and Col IV decellularized group show clear cell nucleus. Please explain why there are cell nucleus in the decellularized groups.
6. In Figure 6a and Figure 4, the legend is not readable. Also the authors should put Figure 4 ahead of Figure 5 and Figure 6.
Author Response
Major issues:
- A Triton x-100 control or SDS group is needed.
Response: Thanks for this point to mention. In fact the main objective of this article is to report the sequence of our previous work, where our group has been published the triton based decellularization (TRICOL) of the cardiac tissues (ref: ACS Biomater Sci Eng. 2020 Oct 12;6(10):5493-5506. doi: 10.1021/acsbiomaterials.0c00540. Heart Vessels. 2016 Nov;31(11):1862-1873. doi: 10.1007/s00380-016-0839-5. PLoS One. 2014 Jun 18;9(6):e99593. doi: 10.1371/journal.pone.0099593). So within this context, we did not present the control from other protocols.
- At page 3 paragraph 2, please added the duration of each treatment. How long was the 1% Tergitol treatment, sodium cholate treatment, peracetic treatment?
Response :The missing information has been incorporated
- Figure 1, the panel label e is missing in the figure. In panel d and e, seems like the author only did paired two group comparison? However, the legend says statistic was done by one-way ANOVA. Please confirm this.
Response: the label has been added to the figure. One way ANOVA with multiple comparisons was performed.
- Figure 2, the scale bar is not readable and I can barely see the structure in the histology images. Please include higher resolution images.
Response: the images are edited accordingly.
- Figure 3, DAPI staining in elastin and Col IV decellularized group show clear cell nucleus. Please explain why there are cell nucleus in the decellularized groups.
Response: thanks for this comment. The efficiency of the decellularization has to be considered primarily on the basis of DNA quantification. In any case some of the DNA is still remained but less than 50 microgram. In this scenario there is a possibility of nuclear presence in the tissues. Fortunately, this is non-significant number of cells compared to the positive samples. to avoid unnecessary imaging editing we kept the images as it is.
- In Figure 6a and Figure 4, the legend is not readable. Also the authors should put Figure 4 ahead of Figure 5 and Figure 6.
Response: the suggested editing has been incorporated
Reviewer 2 Report
The authors demonstrate the use of a new detergent, Tergitol, for the decellularization of porcine pulmonary valves. The results could be of interest for further development of chemical treatment processes for porcine pulmonary valve decellularization and xenotransplantation. The major claims are well supported by data, which were obtained from thorough biomolecular, biomechanical, and morphological analyses.
The major concerns are about the quality of the presentation, which is not sufficient or distract from understanding the content. The details of suggestions for further improvement are:
1. In Figures 1, 2, 3, 4, 5, and 6, the full words for the abbreviations (e.g. PW, LL, MYO, DC, NPW, DPW, NP, DP, UTS, FS, SHG, DECELL, and so on) are not provided in the figure legend or main text, and a lot of guesswork was needed to understand. They all should be clearly stated at least in the earliest figure legend or main text. A thorough review of all abbreviations used in the figures and appropriate corrections as well as provision of the full meanings are highly recommended.
2. Figures are not in numerical order: Figure 4 is placed after Figure 6.
3. In the caption of Figure 4, there is no explanation about the bar graphs, and the font size is too small to read.
4. In Figures 2, 3, and 8, the scale bars are too small to see.
5. In Figure 6a, the font size is too small and the text in the box cannot be read.
6. In Figure 9, the panels are labeled using same letters, like a, b, a, b, c, d: each panel should be labeled using different letters.
7. In Figure 1, the panel name (e) is missing in the image.
8. In Figure 5, the panel name (a) is missing in the image.
9. On Pages 2, 16, and 17, only one or two sentences are written as separate paragraphs, which distracts understanding. Restructuring them into fewer paragraphs is suggested.
Author Response
Dear Prof.
Thank you for the useful revision. please find our responses in the following:
- In Figures 1, 2, 3, 4, 5, and 6, the full words for the abbreviations (e.g. PW, LL, MYO, DC, NPW, DPW, NP, DP, UTS, FS, SHG, DECELL, and so on) are not provided in the figure legend or main text, and a lot of guesswork was needed to understand. They all should be clearly stated at least in the earliest figure legend or main text. A thorough review of all abbreviations used in the figures and appropriate corrections as well as provision of the full meanings are highly recommended.
Response: Thank you very much for this point. The explanation of samples with their abbreviations are added into the text lines 103 to 105. Furthermore, these terms are also explained in the figure captions.
- Figures are not in numerical order: Figure 4 is placed after Figure 6.
Response: The figures are placed in order.
- In the caption of Figure 4, there is no explanation about the bar graphs, and the font size is too small to read.
Response: the suggested editing has been incorporated. The figures size has been increased to make it visible.
- In Figures 2, 3, and 8, the scale bars are too small to see.
Response: the suggested editing has been incorporated
- In Figure 6a, the font size is too small and the text in the box cannot be read.
Response: the suggested editing has been incorporated
- In Figure 9, the panels are labeled using same letters, like a, b, a, b, c, d: each panel should be labeled using different letters.
Response: the figures are edited according to the suggested format.
- In Figure 1, the panel name (e) is missing in the image.
Response: the legend is added to the figure as (e)
- In Figure 5, the panel name (a) is missing in the image.
Response: the legend is added to the figure as (a)
- On Pages 2, 16, and 17, only one or two sentences are written as separate paragraphs, which distracts understanding. Restructuring them into fewer paragraphs is suggested.
Response: Thanks, suggested changes in the text paragraphs has been incorporated.
best regards,
Saima
Reviewer 3 Report
How is this paper different from the previously published paper by the same group (Reference 21 in the article)?
Faggioli, M.; Moro, A.; Butt, S.; Todesco, M.; Sandrin, D.; Borile, G.; Bagno, A.; Fabozzo, A.; Romanato, F.; Marchesan, M.; Imran, S.; Gerosa, G. A New Decellularization Protocol of Porcine Aortic Valves Using Tergitol to Characterize the Scaffold with the Biocompatibility Profile Using Human Bone Marrow Mesenchymal Stem Cells. Polymers 2022, 14, 1226. doi:10.3390/polym14061226.
Author Response
Dear Prof.,
Thanks for the useful suggestions. our response is as follows:
How is this paper different from the previously published paper by the same group (Reference 21 in the article)?
Faggioli, M.; Moro, A.; Butt, S.; Todesco, M.; Sandrin, D.; Borile, G.; Bagno, A.; Fabozzo, A.; Romanato, F.; Marchesan, M.; Imran, S.; Gerosa, G. A New Decellularization Protocol of Porcine Aortic Valves Using Tergitol to Characterize the Scaffold with the Biocompatibility Profile Using Human Bone Marrow Mesenchymal Stem Cells. Polymers 2022, 14, 1226. doi:10.3390/polym14061226.
Response: The main difference between these two research articles ais the tissue type where we are applying the decellularization protocol for the aortic and pulmonary roots in former and later cases. This study is part of our ongoing project to create the bioprostheses or biomaterial which can apply in preclinical and clinical purposes for the cardiac surgery. So study is the being reported as the sequence of the previous study with the application for two different tissues.
Best regards,
Saima
Round 2
Reviewer 1 Report
I can not find the point-by-point response to my comments.
Author Response
Dear Prof.
please accept my apologies as my comments were not visible to you. Now I uploaded the responses after reviewing again the comments and their responses as an attachment. I will be pleased to listen your useful suggestions again to make the article improved.
Best regards,
Saima Imran

Reviewer 3 Report
Thanks for you response! I appreciate the work that the authors must have put in to generate the data, however, the paper does not add enough to the present literature due to the previous publication. The two tissues are very similar, hence the data is nearly identical and so is the structure of the manuscript.
In my opinion, this data along with your ongoing preclinical/clinical work will make a better manuscript and contribution to the field.
Author Response
Dear Prof.
Thank you again for your suggestion. We have been updated the literature. The strength of our data, is the application of new detergent, Tergitol, which is being reported by our group and successfully replaced with the Triton, which was the main hurdle to lead to the preclinical research due to its toxicity and not acceptable for the clinical samples.
We reported this detergent to apply for different cardiac tissue samples. As this research is part of our future preclinical data, so for sure, we are going to report the preclinical part with the higher animals in coming future.
Best regards,
Saima Imran
Round 3
Reviewer 1 Report
the authors addressed all my concerns.
Author Response
Dear Prof.,
Thank you for reviewing the paper submitted by our group entitled: Tergitol based decellularization protocol improves the prerequisites for pulmonary xenografts: characterization and biocompatibility assessment. We have revised the manuscript based on your comments.
Reviewer 3 Report
Same problem persists
Author Response
Dear Prof,
Thank you again for the comments. Your opinion about the in vivo preclinical study is appreciated. However, our preclinical project is based on the implantation of whole pulmonary and aortic roots into the pig hearts with the minimum follow ups of 6 months to 1 year. This project is in realization in couple of months. Before this implantation process, we must have to prove the structural characterization of both the tissues to explore the efficiency of the detergent in the decellularization procedure.
Our group is reporting this detergent for the first time with different tissues, so the background data is not available in most of the cases. Due to the limitation for the self citation data, we did not include most of our publications from the reported data for in vivo implantation studies. following are our reported data from the Triton X-100 based decellularized tissue.
- ACS Biomater Sci Eng. 2020 Oct 12;6(10):5493-5506. doi: 10.1021/acsbiomaterials.0c00540.
- Heart Vessels. 2016 Nov;31(11):1862-1873. doi: 10.1007/s00380-016-0839-5.
- PLoS One. 2014 Jun 18;9(6):e99593. doi: 10.1371/journal.pone.0099593).
I request you to consider this aspect of the present study for submission this format.
Best regards,
Saima